# Positron Emission-Computed Tomography, Cryobiopsy versus Bronchoalveolar Lavage and Computed Tomography Findings for Interstitial Lung Disease: A Long-Term Follow-Up

**DOI:** 10.3390/medicina59040787

**Published:** 2023-04-18

**Authors:** Paul Zarogoulidis, Dimitris Matthaios, Haidong Huang, Chong Bai, Wolfgang Hohenforst-Schmidt, Konstantinos Porpodis, Dimitris Petridis, Konstantinos Pigakis, Nikolaos Kougkas, Panagoula Oikonomou, Christina Nikolaou, Dimitris Hatzibougias, Chrysanthi Sardeli

**Affiliations:** 1Pulmonary Department, General Clinic Euromedica, 54454 Thessaloniki, Greece; 23rd Surgery Department, AHEPA University Hospital, Aristotle University of Thessaloniki, 54636 Thessaloniki, Greece; 3Oncology Department, General Hospital of Rhodes, 85133 Rhodes, Greece; 4Department of Respiratory & Critical Care Medicine, Changhai Hospital, the Second Military Medical University, Shanghai 200433, China; 5Sana Clinic Group Franken, Department of Cardiology/Pulmonology/Intensive Care/Nephrology, “Hof” Clinics, University of Erlangen, 91052 Hof, Germany; 6Pulmonary Department, “George Papanikolaou” General Hospital, Aristotle University of Thessaloniki, 56403 Thessaloniki, Greece; 7Department of Food Technology, School of Food Technology and Nutrition, Alexander Technological Educational Institute, 57400 Thessaloniki, Greece; 8Intensive Care Unit, Creta Interclinic, 71304 Iraklio, Greece; 9Rheumatology Department, Ippokrateio University General Hospital, 54642 Thessaloniki, Greece; 10Second Department of Surgery, University Hospital of Alexandroupolis, Medical School, Democritus University of Thrace, 68100 Alexandroupolis, Greece; 11Private Pathology Lab., Microdiagnostics Ltd., 54622 Thessaloniki, Greece; 12Department of Pharmacology & Clinical Pharmacology, School of Medicine, Aristotle University of Thessaloniki, 54124 Thessaloniki, Greece

**Keywords:** cryobiopsy, ILD, computed tomography scan, positron emission-computed tomography, bronchoscopy

## Abstract

*Background and Objectives:* Interstitial lung diseases have always been an issue for pulmonary and rheumatology physicians. Computed tomography scans with a high-resolution protocol and bronchoalveolar lavage have been used along with biochemical blood tests to reach a diagnosis. *Materials and Methods:* We included 80 patients in total. First, all patients had their diagnosis with computed tomography of the thorax, serological/ immunological blood tests and bronchoalveolar lavage. However; after 3 months, all were divided into 2 groups: those who had bronchoalveolar lavage again and those who had cryobiopsy instead of bronchoalveolar lavage (40/40). Positron emission-computed tomography was also performed upon the first and second diagnosis. The patients’ follow-up was 4 years from diagnosis. *Results:* Patients suffered most from chronic obstructive pulmonary disease (56, 70%), while lung cancer was rarely encountered in the sample (7, 9.75%). Age distribution ranged between 53 and 68 years with a mean value of 60 years. The computed tomography findings revealed 25 patients with typical diagnosis (35.2%), 17 with interstitial pulmonary fibrosis (23.9%) and 11 with probable diagnosis (11%). The cryobiopsy technique led to a new diagnosis in 28 patients (35% of the total sample). Patients who had a new diagnosis with cryobiopsy had a mean survival time of 710 days (<1460). Positron emission-computed tomography SUV uptake was positively associated with the cryobiopsy technique/new disease diagnosis and improved all respiratory functions. Discussion: Positron emission-computed tomography is a tool that can be used along with respiratory functions for disease evaluation. *Conclusions:* Cryobiopsy is a safe tool for patients with interstitial lung disease and can assist in the diagnosis of interstitial lung diseases. The survival of patients was increased in the cryobiopsy group versus only bronchoalveolar lavage for disease diagnosis.

## 1. Introduction

Interstitial lung diseases (ILD) represent a large heterogeneous group of lung disorders and usually have a wide spectrum of overlapping clinical and radiological features. Therefore, there is often a complex diagnostic and therapeutic approach. Obtaining a definitive diagnosis is very difficult and there is usually a long and complex process of diagnostic work-up. In several situations, treatment often starts for symptom relief that sometimes hides the true underlying disease. A careful medical history along with physical examination, high-resolution chest CT (HRCT), cytological profile with bronchoalveolar lavage (BAL) and serological/immunological tests are necessary for a diagnosis. Until recently, HRCT, serological/immunological, pulmonary function tests and bal were the main examinations performed to identify ILDs. However, in recent years, new equipment has been developed and the diagnostic approach changed [1]. Based on recent guidelines, surgical biopsy of the lung is recommended were possible to elucidate the diagnostic profile of the patient [1]. However, an adequate lung tissue sample involves invasive procedures, such as surgery, with possible complications. Therefore, the method of surgical lung biopsy (SLB) is only indicated for those patients suspected of having idiopathic pulmonary fibrosis (IPF) who are at minimal surgical risk. The mortality rate of surgical lung biopsy is 1.5% [2,3]. It has been observed that up to 15% of patients are left without a specific diagnosis due to poor health, comorbidities, and respiratory failure. These patients are defined as unclassifiable interstitial diseases [1,4]. Surgical lung biopsy is currently considered the gold standard for a definitive histological diagnosis in over 90% of cases [2,5,6]. In the case of idiopathic pulmonary fibrosis (IPF), the diagnosis is underestimated, since the typical radiological pattern of IPF is found in only 50% of cases. In the cases where a high-resolution tomography scan is atypical, there is no reliable diagnosis. In search for a less invasive alternative to surgical lung biopsy, transbronchial biopsy (TBB) was proposed. However, this technique provides very small tissue fragments of pulmonary parenchyma (1–3 mm) with a 22G needle and it has limitations due to crushing artifacts. Ocassionally, there is inadequate material to diagnose a usual interstitial pneumonia (UIP) pattern, the histopathological correlate of IPF. The TBB tissue sample has very low sensitivity. The overall diagnostic yield of TBB varies considerably, from 25% to 75%, based upon the type of lung radiological findings and underlying disease [7,8,9,10]. The diagnostic yield can be as high as 80–90% for nonfibrosing ILDs [9,11,12]. TBB has a diagnostic value in cases of sarcoidosis, eosinophilic pneumonia, hypersensitivity pneumonia, organizing pneumonia, microlytiasis, diffuse alveolar damage, amyloidosis, carcinomatous lymphangitis, proteinosis, and infections [7,8,13,14]. Based on the new diagnostic equipment development of radial-endobronchial ultrasound (radial-ebus), novel cryobiopsy probes with 1.1 mm/1.7 mm and additional guidance systems to fluoroscopy, such as Dyna Computed Tomography (Dyna-CT), the transbronchial lung cryobiopsy (TBLC) has increased in everyday clinical work as a promising and safer alternative to surgical lung biopsies (SLB) in the diagnostic approach to fibrosing ILDs [15]. In the case of transbronchial lung cryobiopsy (TBLC), instead of the bioptic forceps, a probe is introduced through the working channel of the bronchoscope (2.8 mm), cooled to a very low temperature (about −80 °C) for a few seconds. The cryo probe freezes a fragment of the pulmonary parenchyma, which remains attached to the probe and is then retrieved [16]. This method enables the retrieval of a larger fragment of lung parenchyma (7–10 mm) from the specific target area of pulmonary infiltrates more suitable and targeted for diagnostic purposes [17]. Several studies have been published regarding the diagnostic yield and safety of transbronchial lung cryobiopsy (TBLC) for the diagnosis of interstitial lung disease (ILD) [15,18,19,20,21]. The results of TBLC are considered an alternative to surgical lung biopsy (SLB); however, there are minor safety concerns mainly due to hemorrhage [17]. The diagnostic methodology changes are based on new published studies and new diagnostic tools. Therefore, patients must be selective based on a multidisciplinary board of experts and physical status.

## 2. Materials and Methods

We included 80 patients from our outpatient cabinets from January 2017 to January 2023, through the investigational review board committee Bioclinic January 2017. Written informed consent was obtained from each patient before study enrollment. All patients had biochemical blood tests for immunological investigation (Appendix A) and high-resolution-computed tomography scan of the thorax with 1 mm slices. Patients were diagnosed according to the European Respiratory Society, American Thoracic Society, and British Society for Rheumatology and American College of Rheumatology. Positron emission tomography-computed tomography (PET–CT) was performed from the authors personal funding since PET–CT is not covered for interstitial lung disease from the health systems involved. Bronchoalveolar lavage (BAL) was conducted under sedation with a fiberoptic bronchoscope upon diagnosis along with pulmonary function tests such as spirometry (forced vital capacity FVC, forced expiratory volume in 1 s FEV1) and diffusing capacity of the lungs for carbon monoxide (DLCO). After 3 months of diagnosis and treatment, we again performed either bal to 40 patients or cryobiopsy. The aim of the study was to assess whether cryobiopsies alone can lead to a new diagnosis and if the new diagnosis increased patient survival. New pulmonary function tests were performed on all patients 1 year after the initial diagnosis. All patients upon initiation of the protocol were fit to undergo bronchoscopic procedures. Cryobiopsies were performed under general anesthesia and a fiberoptic bronchoscope (Olympus, Osaka, Japan) with a Fuji radial-EBUS and fluoroscopy (Siemens C-Arm or Siemens Dyna CT, Hof, Germany) was used with an ERBEII cryo system (Siemens, Hof, Germany). We used the 1.7 mm probe to obtain biopsies hemostatic powder and or balloon blockers in the case of hemorrhage. The methodology of cryobiopsy has been previously described [22]. The median of the largest diameter of the samples was 7 mm (IQR, 6–8 mm). The bal was obtained under mild sedation, 180 mls of saline were introduced in the working channel of the bronchoscope and 80–120 mls returned and sent directly to our laboratory. If it was not possible to return the samples the same day, they were kept in a refrigerator.

An x-ray was performed in the endoscopy suite immediately after the conclusion of the procedure and 1 more 2 h later for pneumothorax investigation. All bal and tissue samples were sent to the same pathology and immunology laboratory. Radial-Ebus, fluoroscopy (C-Arm Siemens) and/or DYNA-CT (Siemens) was used for the guidance of the cryoprobe. The segment for biopsy was selected based on high-resolution-computed tomography (HRCT) of the chest at a multidisciplinary team discussion (MDT). Pneumothorax was observed in 6/40 patients who underwent cryobiopsy. All patients had stopped their anticoagulants for at least 5 days and had their international normalized ratio (INR), prothrombin time (PT) and partial thromboplastin time (APTT) checked the day of the procedure. A total of 6/40 patients had moderate hemorrhage and only 1/40 severe hemorrhage with a 6-day hospitalization in an intensive care unit (ICU) (Figure 1). The bal procedure and evaluation were performed according to previously published guidelines [23].

### Biopsy Specimens

Biopsy specimens were fixed in 10% formalin and embedded in paraffin. Hematoxylin and eosin (H&E) stained slides were reviewed by the standard protocol of the Microdiagnostics Ltd. clinical pathology laboratory, Thessaloniki, Greece; research samples were processed in an identical manner. Clinical specimens were reviewed under standard clinical care and research specimens were reviewed by an expert lung pathologist who was blinded to subject information.

A total of 40 out of 80 patients initially diagnosed with ILD disease (all ILD included in our study are presented in Table 1) had cryobiopsy. The aim was to find causative superior diagnostic criteria by examining (a) cross-tabulated differences between old and new disease diagnosis and (b) a multiple regression of the cryobiopsy performance as a dependent variable against the survival time of patients (<1460 days, up to 1460), change in disease diagnosis (0, 1) and drug administration (0, 1), and change in respiratory functions (FEV1, FVC and DLCO) taken as pair-wise differences per patient. A forward selection of the most important variables preceded the final model and a promising predictive response of cryobiopsy use was attempted. More precisely, each variable was entered into the model only when the p-entry value was statistically significant, that is, lower than the 0.05 probability level of reference. All data were treated via the JMP 17.0 (JMP Statistical Discovery LLC, Cary, NC, USA) statistical software.

## 3. Results

The medical history of patients is shown in sup, split between males (71, 88.75%) and females (9, 11.25%). Most patients suffered from chronic obstructive pulmonary disease (COPD) (56, 70%), while lung cancer (LC) was rarely encountered in the sample (7, 9.75%). Age distribution ranged between 53 and 68 years with a mean value of 60 years (Figure 2). The computed tomography (CT) findings revealed 25 patients with typical diagnosis (35.2%), 17 with idiopathic pulmonary fibrosis (IPF) (23.9%), and 11 with probable diagnosis (11%).

Table 1 describes the list of 25 autoimmune diseases reported in the patients and Figure 3 arranges the array of false changes in diagnosis on either side of those commonly identified.

Most diagnostic changes were encountered in three diseases: mixed connective tissue disease, Langerhans cell histiocytosis (each with 5 counts), and sarcoidosis (6 counts). One disease was never recovered in the new diagnosis list (anti-SRP autoantibody associated interstitial lung disease, no 9). Table 2 shows that sarcoidosis was the most commonly diagnosed in the old list (10 encounters, 12.5%), while the first two aforesaid diseases were more frequently recorded in the new list (9 and 7 counts, 11.25% and 8.75%, respectively).

The cryobiopsy technique was generally described by 28 new disease changes in diagnosis; that is, in 35% of the total sample (Table 3), no changes were recorded in 24 persons (30%), 12 new diagnoses (15%) were exclusively due to the cryobiopsy performance, and 16 diagnoses (20%) were due to its absence.

The cryobiopsy performance was also examined as a response variable resulting from potential effects of the selected parameters in the study: survival time, change in diagnosis and drugs, and respiratory functions (Figure 2). The latter were further rejected by the forward selection of variables as exerting an insignificant effect on cryobiopsy, however; the change in drugs, albeit not significant (*p* = 0.0916), was kept in the analysis due to a highly statistical effect produced when interacting with a change in diagnosis (*p* = 0.0033). The proposed model was statistically significant as compared with the reduced one (includes only the intercept, *p* < 0.0001) and reliable due to low values of AICc and BIC information criteria (92.9 and 104.0 respectively), as revealed by the lack of apparent outliers as the plot of deviance residuals vs. predictive values. The most important variables in the model were the change in diagnosis (*p* = 0.0042) and its interaction term with a change in drugs, and in a lesser effect on the survival time (*p* = 0.0375).

Moving to the intriguing part of the cryobiopsy prediction response, the alteration of diagnosis associated with cryobiopsy was 7.7 times more probable. Those patients who survived up to 1460 days showed 3.4 times higher probability of reacting positively to a cryobiopsy event than patients with a mean survival time of 710 days (<1460). The prediction profiler favors an 84.5% higher probability of cryobiopsy implementation combined with cases involved with an alteration in diagnosis, greater survival, and no change in drugs. The latter is also synergically connected with a change in disease diagnosis, which is a higher percentage of cryobiopsy response (see interaction plot in Appendix A).

## 4. Discussion

Accurate diagnosis is important for management decisions. During the past twenty years, diagnosis of ILDs was based on radiologic, clinical, and serological/immunological blood tests and less on tissue biopsies [24]. To date, there have been no attempts to highlight a relationship between pathologic honeycombing in transbronchial lung cryobiopsy and clinical findings, radiological features, or mortality. Usual interstitial pattern (UIP) pattern has been defined in the literature and in everyday clinical practice through established criteria and morphological features observed only from surgical lung biopsy. In the recently published literature, the diagnostic yield of cryobiopsy in fibrotic ILDs has been directly applied to these criteria to the samples obtained by cryoprobe [15]. The cryobiopsy procedure and sample evaluation have not been standardized, however; the same procedure has been performed by many centers and the samples are handled in the same manner by many centers [1,4,10,11]. Moreover, the morphological features of cryobiopsies have been well associated with radiological findings and associated with clinical findings and disease diagnosis for ILDs [24]. It has been proposed that two to six biopsies are the best number of samples with a minimum of 1.7 mm cryoprobe [13,25]. Moreover, a transbronchial lung biopsy in the middle third of the lung tissue between the large airways and the pleura, about 1 cm from the pleura itself, is the safest and ideal sample site. Usual interstitial pattern changes are usually localized in the subpleural regions of the lower lobes in interstitial pulmonary fibrosis. The limitations of our study were, first, that there was no correlation between the biopsy sample and pneumothorax. Second, none of our patients performed the 6 min walking test. Moreover, it was not investigated whether there was an association between respiratory function values and pneumothorax. Also, the connection of pneumothorax and the physician who performed the biopsies was not investigated. The pneumothorax was alleviated with a chest tube with a median of 3 days of hospitalization. The rest of the patients were discharged 24 h after the procedure. Moderate hemorrhage was observed and only one patient had severe hemorrhage issues, probably due to the smaller biopsy samples taken compared to previous studies [25]. Taking biopsies from the periphery of the lung protects the patient from severe hemorrhage; however, these patients are more prone to pneumothorax. In our case we performed biopsies from different parts of the lung both centrally and from the periphery and, therefore, in combination with smaller biopsies when compared to previous studies, we had less side effects. However, our results indicated that 35% (out 80 patients) of the patients had their diagnosis altered after the cryobiopsy sample was obtained. Our results are in accordance with previous published studies regarding the diagnostic value of cryobiopsy versus bal, TBB, and only CT findings [8,15,22,26]. Moreover; since we used a radial ebus, we were able to visualize the surrounding vessels from the biopsy site and select the parts of the lung very carefully to avoid hemorrhage, as previously described [27,28]. A Fogarty balloon was placed in all biopsies as a precaution measure. Balloon blockers are essential to protect the patient from life-threatening hemorrhage. In our study, we did not have disease exacerbation and our results concur with previous studies [27,28]. The 35% change in diagnosis along with an increase in survival compared to those with only bal and less survival indicates that tissue is “again the issue” for ILDs [26] Pet-CT can be used upon diagnosis as an additional tool for assessing the disease activity upon diagnosis and definitely can be used as a noninvasive tool for re-evaluation of an ILD disease as previously described [29,30]. Our study is one of the first to evaluate a patient upon diagnosis with CTPET–CT and re-evaluate the same patient after a second sample is obtained either with bal or cryobiopsy. The results of the PET–CT findings are 100% associated with the new diagnosis upon resampling and survival. It is our belief, as previously published, that PET–CT could be used in future guidelines for ILDs. Special consideration has to be taken for these patients not to exacerbate the disease, and this can be achieved through evaluating a possible deterioration of the patient respiratory functions and clinical findings. Moreover, we did not include older patients ≥70 to evaluate the safety profile of cryobiopsy for these patients. A major limitation of our study is that we did not describe in detail the effect of pharmacological treatments on the observed results. Furthermore, regional variability influences the results, because environmental factors, ethnicity, cultural habits, and occupational risks are known to be related to the development of interstitial lung diseases. Finally, we presented the prevalence of different multidisciplinary diagnosis of ILDs according to the latest HRCT patterns of UIP, probable UIP, and indeterminate for UIP by the 2018 ATS/ERS/JRS/ALAT guidelines. The computed tomography (CT) findings revealed 25 patients with typical diagnosis (35.2%), 17 with interstitial pulmonary fibrosis (IPF) (23.9%) and 11 with probable diagnosis (11%). Table 1 describes the list of 25 autoimmune diseases reported in the patients along with the Appendix A and presents an array of false differentiation in the diagnoses on either side of those commonly identified. Most diagnostic changes were encountered in three diseases: mixed connective tissue disease, Langerhans cell histiocytosis (each with 5 counts), and sarcoidosis (6 counts). One disease was never recovered in the new list of diagnoses (anti-SRP autoantibody associated interstitial lung disease). It has been reported that a low reproducibility is that the diagnostic accuracy of the procedure is highly dependent on the location of the biopsies, the size of the specimens, and the degree of patchy inhomogeneity of pathological changes in the lungs. According to many studies, cryobiopsy may appear to be less accurate than surgical biopsy; nevertheless, it is considered a safer method, especially for patients at high surgical risk.

## 5. Conclusions

In general, cryobiopsies are safe, necessary, and should be performed in centers with experience not only from the part of the endoscopists, but also from the pathology department. Surgical lung biopsy still remains the gold standard. A multidisciplinary approach is necessary for these patients; radiologists, pathologists, pulmonary physicians, thoracic surgeons, and rheumatologists should be included in the multidisciplinary board (MDB). Allowing an accurate histological diagnosis in a much wider number of patients, we can offer new treatment options, such as the new antifibrotics, to patients with ILDs who would not otherwise be diagnosed.

## Figures and Tables

**Figure 1 medicina-59-00787-f001:**
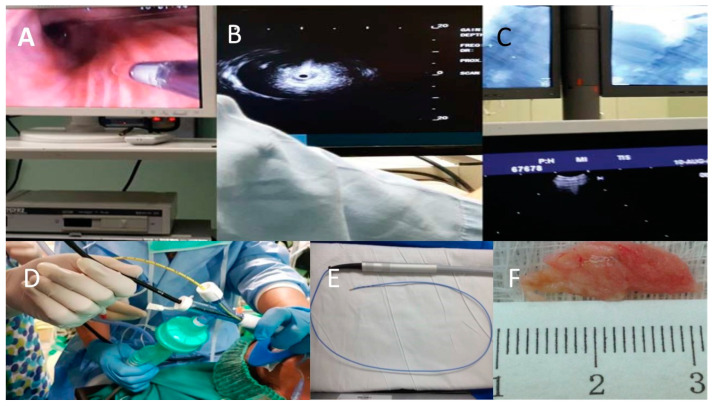
(**A**): radial-ebus probe endobronchially; (**B**): radial-ebus probe ultrasound image; (**C**): fluoroscopy. (**D**): patient intubated with double lumen endotracheal tube with inlet for the endoscope and fogarty balloon; (**E**): ERBE cryoprobe; (**F**): lung parenchyma tissue sample.

**Figure 2 medicina-59-00787-f002:**
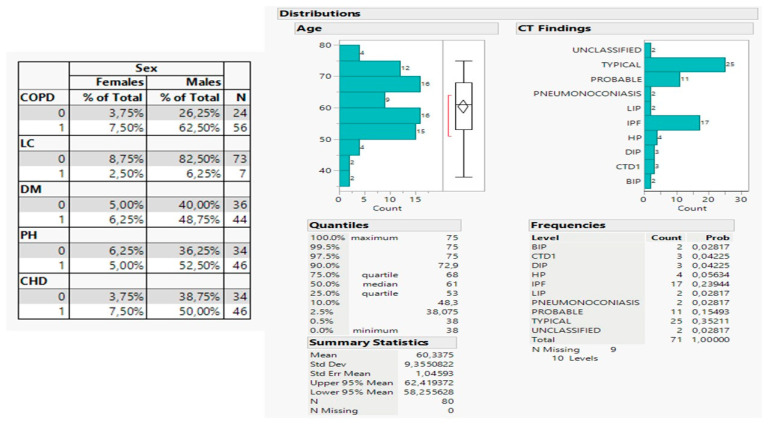
Table of medical history plus age and CT readings distribution. Left: disease distribution, Right: Distribution of age with computed tomography findings. IPF: idiopathic pulmonary fibrosis, LIP: lymphoid interstitial pneumonia, HP: hypersensitivity pneumonia, BIP: bronchiolitis obliterans pneumonia, CTD1: connective tissue disease, DIP: desquamative interstitial pneumonia, COPD: chronic obstructive pulmonary disease, LC: lung cancer, PH: pumonary hypertension, CHD: coronary heart disease, DM: diabetes melitus.

**Figure 3 medicina-59-00787-f003:**
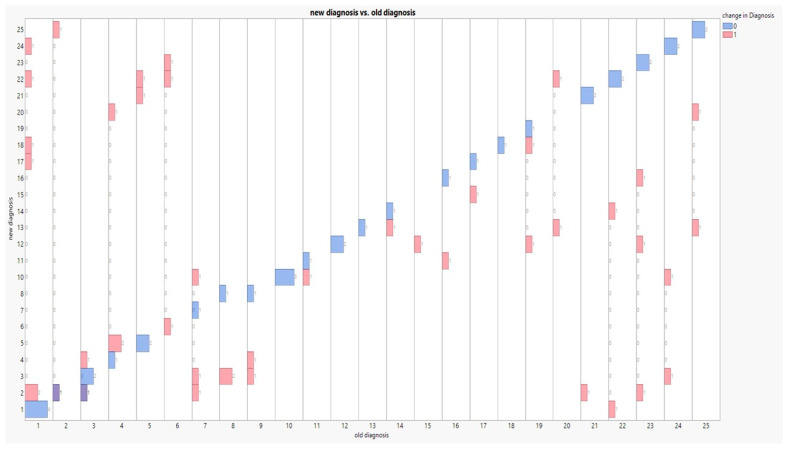
Graphical cross-tabulation between old and new diagnoses as affected by change in diagnosis (0.1). Red bars: changes; Blue bars: no changes.

**Table 1 medicina-59-00787-t001:** List of well-known autoimmune diseases.

1	Sarcoidosis
2	Langerhans cell histiocytosis
3	Vasculitis
4	Systemic lupus erythematosus
5	Lymphangitis carcinomatosis
6	Rheumatoid arthritis
7	Progressive systemic sclerosis
8	Anti-Jo-1 antibody positive interstitial lung disease
9	Anti-SRP autoantibody associated interstitial lung disease
10	Ankylosing spondylitis
11	Sjögren syndrome
12	Mixed connective tissue disease
13	Psoriasis—pulmonary manifestations of psoriasis
14	Interstitial pneumonia with autoimmune features
15	HIV associated interstitial lung diseases
16	Amyloidosis
17	Alveolar proteinosis
18	Idiopathic pulmonary fibrosis
19	Cryptogenic organizing pneumonia
20	Respiratory bronchiolitis-interstitial lung disease
21	Desquamative interstitial pneumonia
22	Lymphoid interstitial pneumonia
23	Acute interstitial pneumonitis
24	Combined pulmonary fibrosis and emphysema
25	Idiopathic pleuroparenchymal fibroelastosis

**Table 2 medicina-59-00787-t002:** A tally of autoimmune disease counts in both diagnoses, old and new.

Disease	Old	Prob	New	Prob
1	10	0.125	5	0.0625
2	3	0.0375	9	0.1125
3	5	0.0625	7	0.0875
4	4	0.05	3	0.0375
5	4	0.05	4	0.05
6	3	0.0375	1	0.0125
7	4	0.05	1	0.0125
8	3	0.0375	2	0.025
9	3	0.0375		
10	3	0.0375	6	0.075
11	2	0.025	2	0.025
12	2	0.025	5	0.0625
13	1	0.0125	4	0.05
14	2	0.025	2	0.025
15	1	0.0125	1	0.0125
16	2	0.025	2	0.025
17	2	0.025	2	0.025
18	1	0.0125	3	0.0375
19	3	0.0375	1	0.0125
20	2	0.025	2	0.025
21	3	0.0375	3	0.0375
22	4	0.05	6	0.075
23	5	0.0625	3	0.0375
24	4	0.05	3	0.0375
25	4	0.05	3	0.0375
Total	80	1	80	1

**Table 3 medicina-59-00787-t003:** Association between cryobiopsy and disease.

Change in Diagnosis	Cryobiopsy	% of Total	*n*
0	0	30	24
	1	15	12
1	0	20	16
	1	35	28

Number 0 indicates the number of patients who had no alteration between the first and second diagnosis with the cryobiopsy technique. Number 1 indicates the number of patients who had a new diagnosis after cryobiopsy was performed.

## Data Availability

Any data can be provided by the corresponding author if requested.

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
