# Peer review of "Positron Emission-Computed Tomography, Cryobiopsy versus Bronchoalveolar Lavage and Computed Tomography Findings for Interstitial Lung Disease: A Long-Term Follow-Up"

_medicina, 2023, doi:10.3390/medicina59040787_

Round 1
Reviewer 1 Report
The authors include 80 patients with ILD based on CTS, serological and immunological tests and bal analysis. After that the authors performed bal analysy in 40 patients and cryobiopsy in 40 patients, as well as CT. They found that most of the patients had the diagnosis of COPD, and less lung cancer. The CT findings revealed 25 patients with typical diagnosis, 17 with IPF and 11 with probable diagnosis. The cryobiopsy analysis caused the change of the diagnosis in 28 patients. The authors suggest that cryobiopsy can significantly assist in diagnosis of ILD, it is safe for patients and increased the survival of patients.
The study is interesting and can have value in diagnostic procedure but huge correction of manuscript should be done.
Major revision:
The diagnosis based on cryobiopsy caused change in diagnosis of 28 patients, which should be expressed (%) relative to patients who underwent cryobiopsy diagnostic procedure.
The current lack of diagnostic procedure regarding ILD should be clearly described in the Introduction section.
The aims of the study are not clearly stated.
The manuscript is full with errors and needs serious English corrections.
What are the conclusions of the study and recommendations regarding diagnostic procedure.
Introduction, last paragraph is very confusing, it should be clearly written and corrected.
All bal and tissue samples were sent to the same pathology and immunology laboratory.
The sentence is not meaningful, the type of analysis performed should be clearly given.
Cryobiopsi and bal procedure should be given briefly
Figure 1, panels A, B, C.. should be marked and explained in Figure legend. All pictures shown on the Figure should be necessary and explained in the text?
All Tables should be uniform.
In methods Biopsy specimen is very confusing given, criterial of the diagnosis of disease are not clearly explained.
Explain in the text what is given in Table one, the connection with your research.
It is not given how the statistics was done, what method was used?
Figure 2 can not be Table.
Figure 3 legend is very confusing.
Primarily given diagnosis should be clearly stated, and the diagnosis after application of new methods.
Table 3. what is cryobiopsy effect?
Some Minor revision:
Page 2 line 54-56, 59-61, 83 syntax errors,
62-63 sentence not clear, need to be corrected
69-70 no new paragraph
71 correlate should be characteristic
The sensitivity is very low. Of what?
90-91 syntax error, sentence is not clear.
Page 5 centance 154-156 is very confusing.
Author Response
Reviewer 1
The authors include 80 patients with ILD based on CTS, serological and immunological tests and bal analysis. After that the authors performed bal analysy in 40 patients and cryobiopsy in 40 patients, as well as CT. They found that most of the patients had the diagnosis of COPD, and less lung cancer. The CT findings revealed 25 patients with typical diagnosis, 17 with IPF and 11 with probable diagnosis. The cryobiopsy analysis caused the change of the diagnosis in 28 patients. The authors suggest that cryobiopsy can significantly assist in diagnosis of ILD, it is safe for patients and increased the survival of patients.
The study is interesting and can have value in diagnostic procedure but huge correction of manuscript should be done.
Major revision:
The diagnosis based on cryobiopsy caused change in diagnosis of 28 patients, which should be expressed (%) relative to patients who underwent cryobiopsy diagnostic procedure.
Thank you for your comment
This information is already in the text in the results section as 35% of the total patient sample. Or if you want 70% of the 40 patient that underwent cryobiopsy.
The current lack of diagnostic procedure regarding ILD should be clearly described in the Introduction section.
Thank you for your comment
Answer
According to ERS, ATS guidelines there is no lack of diagnosis, The diagnostic tools and methodology changes according to the consensus meetings of the ERS and ATS more or less every 2 years. This has to do with new studies published and new diagnostic tools. The following sentence has been added in the introduction section
The diagnostic methodology changes based on new published studies and new diagnostic tools.
The aims of the study are not clearly stated.
Thank you for your comment
Answer
The following sentence has been added in the methods section
The main aim of the study was to assess whether cryobiopsies alone can change the patients` diagnosis and if the new diagnosis increased the patients` survival.
The manuscript is full with errors and needs serious English corrections.
Thank you for your comment
Answer
We have corrected the text both linguistically and grammatically
What are the conclusions of the study and recommendations regarding diagnostic procedure.
Thank you for your comment
Answer
As stated in the discussion section cryobiopsies are safe and the diagnostic yield is higher than that of BAL since they provide tissue. They should be performed where possible based on the clinical status of the patient. PET-CT should be considered as a staging tool, when more clinical trials are available. The procedure of cryobiopsy should be performed under jet-ventilation and sedation only in centers with experience.
Introduction, last paragraph is very confusing, it should be clearly written and corrected.
Thank you for your comment
Answer
We made corrections-additions regarding TBLC and SLB
All bal and tissue samples were sent to the same pathology and immunology laboratory.
The sentence is not meaningful, the type of analysis performed should be clearly given.
Thank you for your comment
Answer
Yes, they were sent in the same laboratory, we mention this in the methods section the lab is Microdiagnostics, Thessaloniki, Greece
The type of tissue analysis is mentioned in the methods section and again it is mentioned in the methods that both tissue and bal samples were handled according to published guidelines references 22, 23
Number
Kronborg-White, S.; Folkersen, B.; Rasmussen, T.R.; Voldby, N.; Madsen, L.B.; Rasmussen, F.; Poletti, V.; Bendstrup, E. Introduction of cryobiopsies in the diagnostics of interstitial lung diseases - experiences in a referral center. European clinical respiratory journal 2017, 4, 1274099, doi:10.1080/20018525.2016.1274099.
Meyer, K.C.; Raghu, G.; Baughman, R.P.; Brown, K.K.; Costabel, U.; du Bois, R.M.; Drent, M.; Haslam, P.L.; Kim, D.S.; Nagai, S.; et al. An official American Thoracic Society clinical practice guideline: the clinical utility of bronchoalveolar lavage cellular analysis in interstitial lung disease. American journal of respiratory and critical care medicine 2012, 185, 1004-1014, doi:10.1164/rccm.201202-0320ST.
Cryobiopsi and bal procedure should be given briefly
The cryobiopsy procedure is already mentioned in the methods section, and the bal procedure has now been added and highlighted in yellow.
The bal was obtained under mild sedation, 180mls of saline were introduced in the working channel of the bronchoscope and 80-120mls returned and sent directly to our laboratory, if not possible the same day, then they were kept in a refrigerator.
Figure 1, panels A, B, C.. should be marked and explained in Figure legend. All pictures shown on the Figure should be necessary and explained in the text?
Thank you for your comment
Answer
The figures are explained in details, in the legend, the procedure of cryobiopsy is mentioned in reference number 22
Kronborg-White, S.; Folkersen, B.; Rasmussen, T.R.; Voldby, N.; Madsen, L.B.; Rasmussen, F.; Poletti, V.; Bendstrup, E. Introduction of cryobiopsies in the diagnostics of interstitial lung diseases - experiences in a referral center. European clinical respiratory journal 2017, 4, 1274099, doi:10.1080/20018525.2016.1274099.
It is not a new method and there is no need to explain it again further, we did not do anything new on this part. Also, the procedure is presented, not in extent in the methods section.
All Tables should be uniform.
Thank you for your comment
Answer
They are
In methods Biopsy specimen is very confusing given, criterial of the diagnosis of disease are not clearly explained.
Thank you for your comment
Answer
We have mentioned in themethods section that diagnosis for the patients was according to Patients were diagnosed according to European Respiraory Society, American Thoracic Society and British Society for Rheumatology and American College of Rheumatology.
Explain in the text what is given in Table one, the connection with your research.
Thank you for your comment
Answer
We have added the sentence (all ILD included in our study are presented in table 1) in the methods section
It is not given how the statistics was done, what method was used?
Thank you for your comment
Answer
We disagree the methodology is cleary stated, however; we have now added in the methods section the following: . More precisely, each variable was entered into the model only when the p-entry value was statistically significant, that is lower than the 0.05 probability level of reference. All data were treated via the JMP 17.0 (JMP Statistical Recovery LLC, 2022) statistical software.
Figure 2 can not be Table.
Thank you for your comment
Answer
It is not considered a Table, there are figures within it
Figure 3 legend is very confusing.
Thank you for your comment
Answer
We are sorry for this, however; they are statistics terms
Primarily given diagnosis should be clearly stated, and the diagnosis after application of new methods.
Thank you for your comment
Answer
This information is already included in the results section
Table 3. what is cryobiopsy effect?
Thank you for your comment
Answer
It means the change of the initial diagnosis to the final diagnosis
Some Minor revision:
Page 2 line 54-56, 59-61, 83 syntax errors,
Thank you for your comment
Answer
Linguistically corrections have been made
62-63 sentence not clear, need to be corrected
Thank you for your comment
Answer
Linguistically corrections have been made
69-70 no new paragraph
Thank you for your comment
Answer
Linguistically and grammatical corrections have been made
71 correlate should be characteristic
The sensitivity is very low. Of what?
Thank you for your comment
Answer
Linguistically corrections have been made
90-91 syntax error, sentence is not clear.
Thank you for your comment
Answer
Linguistically corrections have been made
Page 5 centance 154-156 is very confusing.
Thank you for your comment
Answer
Linguistically corrections have been made

Reviewer 2 Report
The work deals with a very interesting topic. It is written correctly. The presented conclusions result from the presented results. I would recommend a brief description of the statistical tools used in the analysis of the obtained results.
Author Response
Reviewer 2
The work deals with a very interesting topic. It is written correctly. The presented conclusions result from the presented results. I would recommend a brief description of the statistical tools used in the analysis of the obtained results.
Thank you for your comment
Answer
The following information have been added and highlighted in yellow in the METHODS section.
A forward selection of the most important variables preceded the final model and a promising predictive response of cryobiopsy use was attempted. More precisely, each variable was entered into the model only when the p-entry value was statistically significant, that is lower than the 0.05 probability level of reference. All data were treated via the JMP 17.0 (JMP Statistical Recovery LLC, 2022) statistical software.

Round 2
Reviewer 1 Report
Article: PET-CT, Cryobiopsy Versus Bal and CT findings for ILD: A Long Term
Follow up.
The authors added corrections in the manuscript, however still improvements of the text are necessary. One of the most important things is that authors should do correction of English language or English editing by professional English editor from the medical field!
Abbreviations should be avoided in the title of the manuscript and full names should be given.
The cryobiopsy technique changed the diagnosis in 28 patients (35% of the total sample).
Technique can not change diagnosis. Application of technique caused/led to establishment/change/alteration of other/new diagnosis….(change wherever)
Can significantly assist, it is not good expression, …have significant effect on, or assist in…though the meaning is not the same.
The main aim…it is the aim of the study, there are no other aims mentioned….crypbiopsy can not change patient diagnosis , it can affect or influence or cause establishing of new diagnosis
Figure 2. Firstly contains table which should be given as Table. Than the authors should draw their own histogram figures not to use statistical output figures, as well as generate tables not to present Figure with the Table from statistical output. Use the same Font as in the manuscript in the Tables.
They showed figure with the table, not the table, and Table should be Table !
Correct uniformly within whole manuscript.
Why different length of number for p is used in Figures/Tables
P is non significan or ns, or p<0.05, p<0.01, p<0.001……so many numbers looks messy
Why different length of number for p is used in Figures/Tables
P is non significan or ns, or p<0.05, p<0.01, p<0.001……so many numbers looks messy.
Cryobiopsy effect, cryobiopsy is tissue sample taken and frozen, what can be its effect?
There is no cryobiopsy effect, it should be corrected that it is meaningful.
Author Response
The authors added corrections in the manuscript, however still improvements of the text are necessary. One of the most important things is that authors should do correction of English language or English editing by professional English editor from the medical field!
Thank you for your comments
Answer
We do not have budget for language editing. Therefore we requested from Dr. J. Francis Turner University of Tennessee Graduate School of Medicine, Department of Medicine, Knoxville, TN, USA. An expert pulmonary physician with expertise in interventional pulmonology to make additional grammatical and linguistical corrections to the text.
Abbreviations should be avoided in the title of the manuscript and full names should be given.
Thank you for your comments
Answer
We changed the title as you requested
Positron Emission Computed Tomograpgy, Cryobiopsy Versus Bronchoalveolar Lavage and Computed Tomography findings for Interstitial Lung Disease: A Long Term Follow up
The cryobiopsy technique changed the diagnosis in 28 patients (35% of the total sample).
Technique can not change diagnosis. Application of technique caused/led to establishment/change/alteration of other/new diagnosis….(change wherever)
Thank you for your comments
Answer
We made the necessary changes were necessary and we highlighted these changes in yellow
Can significantly assist, it is not good expression, …have significant effect on, or assist in…though the meaning is not the same.
Thank you for your comments
Answer
We made the necessary changes were necessary and we highlighted these changes in yellow
The main aim…it is the aim of the study, there are no other aims mentioned….crypbiopsy can not change patient diagnosis , it can affect or influence or cause establishing of new diagnosis
Thank you for your comments
Answer
We made the necessary changes were necessary and we highlighted these changes in yellow
Figure 2. Firstly contains table which should be given as Table. Than the authors should draw their own histogram figures not to use statistical output figures, as well as generate tables not to present Figure with the Table from statistical output. Use the same Font as in the manuscript in the Tables.
They showed figure with the table, not the table, and Table should be Table !
Correct uniformly within whole manuscript.
Thank you for your comments
Answer
The authors disagree, we want to present our data as we have
Why different length of number for p is used in Figures/Tables
P is non significan or ns, or p<0.05, p<0.01, p<0.001……so many numbers looks messy
Thank you for your comments
Answer
We have explained thoroughly our statistical methodology, we cannot provide any additional information
Why different length of number for p is used in Figures/Tables
P is non significan or ns, or p<0.05, p<0.01, p<0.001……so many numbers looks messy.
Thank you for your comments
Answer
We have explained thoroughly our statistical methodology, we cannot provide any additional information
Cryobiopsy effect, cryobiopsy is tissue sample taken and frozen, what can be its effect?
There is no cryobiopsy effect, it should be corrected that it is meaningful.
Thank you for your comments
Answer
We use the word ``effect`` metaphorically, adding to cryobiopsy methodology the ability to change the diagnosis. We added ``effect`` throughout the text.

Round 3
Author Response
Reviewer 1
The correction of English language is not sufficient, the authors should correct English language of the manuscript.
Thank you for your comment
Answer
We have corrected the manuscript again
Title should be:
Positron Emission Computed Tomograpgy and Cryobiopsy Versus
Bronchoalveolar Lavage and Computed Tomography in diagnostic of Interstitial Lung Disease: A Long Term Follow up
Thank you for your comment
Answer
We have changed the title as you wanted
Abbreviations are not introduced in the manuscript properly
Thank you for your comment
Answer
We have corrected where necessary
Technique can not change diagnosis. 38
Thank you for your comment
Answer
We have corrected where necessary
The cryobiopsy technique led to the establishment of a new diagnosis
39
Thank you for your comment
Answer
We have corrected where necessary
Patients with a new diagnosis had a mean survival time of 710 days (<1460).
43
and can assist in the diagnostic of ILDs
Figure 2. contains picture of table, necessary results should be presented in Table, and histograms should be drawn. This is not acceptable.
Thank you for your comment
Answer
We have removed figure number 2, and placed it in the supplementary data
The effect of using cryobiopsy technique was change in diagnosis….
Table 3 title is very confusing, what is presented in the first column?
Thank you for your comment
Answer
We have changed the title of table 3 to:
Table 3. Association between cryobiopsy and disease. Number 0 indicates the number of patients that had no alteration between the first and second diagnosis with cryobiopsy technique. Number 1 indicates the number of patients that had a new diagnosis after cryobiopsy was performed.
The authors should make the tables where clearly would be stated what is presented in the table not to show pictures of statistical outputs.
Change in diagnosis of ILD after applying cryobiopsy technique.
This is not literature or philosophical work to use metaphoric expressions, it is scientific work and results should be presented and discussed in a clear and precise manner in order to be understandable for readers.
Thank you for your comment
Answer
We have removed figure number 2 and also we have corrected the title in Table number 3
